# Octadecaneuropeptide, ODN, Promotes Cell Survival against 6-OHDA-Induced Oxidative Stress and Apoptosis by Modulating the Expression of miR-34b, miR-29a, and miR-21in Cultured Astrocytes

**DOI:** 10.3390/cells13141188

**Published:** 2024-07-12

**Authors:** Amine Bourzam, Yosra Hamdi, Seyma Bahdoudi, Karthi Duraisamy, Mouna El Mehdi, Magali Basille-Dugay, Omayma Dlimi, Maher Kharrat, Anne Vejux, Gérard Lizard, Taoufik Ghrairi, Benjamin Lefranc, David Vaudry, Jean A. Boutin, Jérôme Leprince, Olfa Masmoudi-Kouki

**Affiliations:** 1Laboratory of Neuroendocrine, Endocrine and Germinal Differentiation and Communication (NorDiC), Inserm UMR 1239, University Rouen Normandie, 76000 Rouen, France; bourzamamine77@gmail.com (A.B.); karthi.duraisamy@univ-rouen.fr (K.D.); mouna.elmehdi@sf.mpg.de (M.E.M.); magali.basille@univ-rouen.fr (M.B.-D.); omayma.dlimi1@univ-rouen.fr (O.D.); benjamin.lefranc@univ-rouen.fr (B.L.); david.vaudry@univ-rouen.fr (D.V.); ja.boutin.pro@gmail.com (J.A.B.); jerome.leprince@univ-rouen.fr (J.L.); 2LR18ES03 Laboratory of Neurophysiology, Cellular Physiopathology and Valorisation of Biomolecules, Faculty of Science of Tunis, University Tunis El Manar, Tunis 2092, Tunisia; h.yosra@yahoo.fr (Y.H.); seyma.bahdoudi@usherbrooke.ca (S.B.); taoufik.ghrairi@fst.utm.tn (T.G.); 3Human Genetics Laboratory (LR99ES10), Faculty of Medicine of Tunis, University of Tunis El Manar, Tunis 2092, Tunisia; maher.kharrat@fmt.utm.tn; 4Centre des Sciences du Goût et de l’Alimentation (CSGA), CNRS, INRAE, Institut Agro, Université de Bourgogne, 21000 Dijon, France; anne.vejux@u-bourgogne.fr; 5Team Bio-PeroxIL, “Biochemistry of the Peroxisome, Inflammation and Lipid Metabolism” (EA7270), Université de Bourgogne, Inserm, 21000 Dijon, France; gerard.lizard@u-bourgogne.fr

**Keywords:** ODN, 6-OHDA, Parkinson’s disease, miR, neuroprotection, apoptosis

## Abstract

Astrocytes specifically synthesize and release endozepines, a family of regulatory peptides including octadecaneuropeptide (ODN). We have previously reported that ODN rescues neurons and astrocytes from 6-OHDA-induced oxidative stress and cell death. The purpose of this study was to examine the potential implication of miR-34b, miR-29a, and miR-21 in the protective activity of ODN on 6-OHDA-induced oxidative stress and cell death in cultured rat astrocytes. Flow cytometry analysis showed that 6-OHDA increased the number of early apoptotic and apoptotic dead cells while treatment with the subnanomolar dose of ODN significantly reduced the number of apoptotic cells induced by 6-OHDA. 6-OHDA-treated astrocytes exhibited the over-expression of miR-21 (+118%) associated with a knockdown of miR-34b (−61%) and miR-29a (−49%). Co-treatment of astrocytes with ODN blocked the 6-OHDA-stimulated production of ROS and NO and stimulation of *Bax* and *caspase-3* gene transcription. Concomitantly, ODN down-regulated the expression of miR-34b and miR-29a and rescued the 6-OHDA-associated reduced expression of miR21, indicating that ODN regulates their expression during cell death. Transfection with miR-21-3p inhibitor prevented the effect of 6-OHDA against cell death. In conclusion, our study indicated that (i) the expression of miRNAs miR-34b, miR-29a, and miR-21 is modified in astrocytes under 6-OHDA injury and (ii) that ODN prevents this deregulation to induce its neuroprotective action. The present study identified miR-21 as an emerging candidate and as a promising pharmacological target that opens new neuroprotective therapeutic strategies in neurodegenerative diseases, especially in Parkinson’s disease.

## 1. Introduction

Parkinson’s disease (PD) is mainly characterized by the loss of dopaminergic neurons in the mesencephalic substantia nigra pars compacta (SNpc), which project toward the striatum. This neuronal degeneration leads to a dopamine deficit in the striatum, which is responsible for the motor symptoms (resting trembles, muscular hypertonia, akinesia, and bradykinesia) encountered by Parkinson’s patients. The development of PD is the result of the interaction between genetic factors [1] and environmental neurotoxins [2,3], which are responsible for a marked oxidative stress and mitochondrial respiratory chain dysfunction, leading to the apoptotic death of dopaminergic nigrostriatal neurons and the development of a neuroinflammatory response that promotes neuronal loss and the progression of the disease.

Current therapies for PD fail to prevent disease progression and the efficacy of these treatments declines over time, highlighting the importance of developing therapies based on neuroprotection and neuro-restoration. Neuroprotection therapy consists of creating and/or modifying the pathophysiological environment of the nigrostriatal region into an environment that preserves the functionality of dopaminergic neurons [4]. Neuro-restoration may involve the repopulation of dopaminergic neurons by transplanting fetal dopaminergic neurons or endogenous neuroprogenitor cells [5,6]. A neuroprotective approach is to identify neurotrophic and/or neuroprotective factors [4,7] that are capable of arresting or limiting processes leading to cell death such as oxidative stress and neuroinflammation. Because of their ability to act directly on the apoptotic mechanisms of dopaminergic neurons in the SNpc and insight into their mode of action, endogenous neurotrophic factors represent serious candidates for the development of neuroprotective strategies [4,8,9,10]. Indeed, some endogenous neuroprotective neuropeptides, i.e., glial cell line-derived neurotrophic factor and activity-dependent neurotrophic factor (ADNF), are already in use in clinical trials [7,10].

Astrocytes play an important role in supporting and maintaining the functions of neurons [11]. It is most important to identify new factors expressed by astrocytes that can prevent both oxidative stress and the neuroinflammation responsible for apoptosis-induced neuron degeneration in PD. In this context, octadecaneuropeptide (ODN), which is a neurotrophic factor, is exclusively produced by astrocytes in the central nervous system (CNS) in mammals [12] and rescues neurons and glial cells from neurotoxicity induced by several substances such as hydrogen peroxide (H_2_O_2_) [13], 6-hydroxydopamine (6-OHDA) [14,15], and 1-methyl-4-phenyl-1,2,3,6-tetrahydropyridine (MPTP) [16]. Thus, ODN could be a promising therapeutic candidate. ODN is a peptide generated through the proteolytic cleavage of diazepam-binding inhibitor (DBI), a precursor of endozepine-family peptides [12,17]. ODN acts through two types of receptors, the former central-type benzodiazepine receptors (CBR) associated with the GABA-A receptor complex [18,19] and a still-unknown G protein-coupled receptor (GPCR) positively coupled either to phospholipase C [20,21,22,23] or adenylyl cyclase [23] and MAPK-ERK transduction pathways [14,24].

There is evidence that endozepine production is induced during injuries involving oxidative neurodegeneration such as inflammation and neurodegenerative diseases. Clinical studies have shown that endozepine, ODN, and DBI levels are increased in the cerebrospinal fluid in patients suffering from neurological disorders including PD [12,25]. Mass spectrometry analyses have also revealed that moderate oxidative stress increases the release of ODN from cultured astrocytes [26]. In fact, the deficiency of ODN due to *DBI* gene knockout results in both an exacerbation of oxidative assaults in neuronal cultured cells [26] as well as the increased sensitivity of dopaminergic neurons in the SNpc and ventral tegmental area to MPTP neurotoxicity in PD mouse model [16]. These data imply that ODN may be an endogenous peptide with neuroprotective properties preventing neuronal cell death.

MicroRNAs (miRNAs) are small non-coding RNAs that regulate gene expression by targeting messenger RNA (mRNA) and inhibiting protein translation [27,28]. MiRNAs have been implicated as regulators of various cellular and physiological processes such as development, the control of cell growth, differentiation, proliferation, and a variety of other biological processes [29]. Moreover, miRNAs have also emerged as key mediators in the etiology of a variety of pathological processes such as neurological disorders including PD [30,31]. Specifically, miRNAs identified in astrocytes have been shown to play a crucial role in the regulation of gene expression in response to PD [32]. Among all miRNAs, miR-21, miR-29a, and miR-34b, expressed notably by astroglial cells, have been shown to be involved in the pathophysiology of PD [33,34,35,36] by regulating cytokine production, inflammatory responses, oxidative stress, and cell apoptosis [33,37].

Although there is clear evidence that ODN exerts a strong protective activity against 6-OHDA-induced apoptosis on cultured neurons and glial cells, a role for miRNAs in the neuroprotective effects of ODN is not well known and has not been shown previously. The purpose of the present study was to examine the potential protective action of ODN when used after the occurrence of insults induced by 6-OHDA in cultured astrocytes and to investigate the effects of the peptide in expression levels of miR-21, miR-29a, and miR-34b.

## 2. Materials and Methods

### 2.1. Animals

Adult Wistar rats were purchased from Pasteur Institute of Tunis (Tunis, Tunisia) or provided by the Biological Resources Service at HeRacLeS research infrastructure, Inserm US 51, CNRS UAR 202 (Rouen, France), maintained in a temperature-controlled room (22 ± 2 °C) under a 12 h light-dark photoperiod cycle and with free access to food and water. The approval of these experiments was obtained from the Medical Ethics Committee for the Care and Use of Laboratory Animals of the Pasteur Institute of Tunis (approval number: FST/LNFP/Pro152012) and our Institutional Animal Use and Care Committee (CENOMAX, Agreement of the Ministry of Research n°54, Agreement n°: # 2021020814212189).

### 2.2. Chemicals

Dulbecco’s modified Eagle’s medium (DMEM), D(+)-glucose, F-12, L-glutamine, fetal bovine serum (FBS), N-2-hydroxyethylpiperazine-N-2-ethane sulfonic acid (HEPES) buffer solution, antibiotic–antimycotic solution, and trypsin-EDTA were purchased from Gibco (Invitrogen, Grand Island, NY, USA). 6-OHDA, fluorescein diacetate–acetoxymethyl ester (FDA-AM), bovine serum albumin (BSA), lactate dehydrogenase (LDH) assay kit, and insulin were obtained from Sigma-Aldrich (St. Louis, MO, USA). Dimethylsulfoxide (DMSO), 2′,7′-dichlorodihydrofluorescein diacetate, and acetyl ester (H_2_DCFDA) were from Invitrogen; nitrogen monoxide dye 4-amino-5-methylamino-2′,7′-difluorofluorecein diacetate (DAF-FM DA) was from Molecular Probs (Eugen, OR, USA). Flumazenil and barbadin were obtained from MedChemExpress (Nonmouth Junction, NJ, USA). Rat ODN (QATVGDVNTDRPGLLDLK) was synthesized by using the standard fluorenylmethyloxycarbonyl (Fmoc) procedure as previously described [21].

### 2.3. Cell Culture

Secondary cultures of astrocytes were prepared from 1- or 2-day-old Wistar rats of both sexes as previously described [38] with minor modifications. Briefly, cerebral hemispheres from 1- or 2-day-old Wistar rats of both sexes were collected in DMEM/F12 culture medium (2:1; *v*/*v*) supplemented with 2 mM glutamine, 1 ‰ insulin, 5 mM HEPES, 0.4% glucose, and 1% of the antibiotic–antimycotic solution (Gibco, Thermo Fisher Scientific Inc., Waltham, MA, USA—Ref. 15240062). Tissues were mechanically dissociated with a syringe fitted with a 1 mm needle and filtered through a 100 μm sieve. Dissociated cells were suspended in culture medium supplemented with 10% FBS, plated in 75 cm^2^ (Greiner Bio-One GmbH, Frickenhausen, Germany) at a density of 17 × 10^6^ cells/mL, and incubated at 37 °C in a 5% CO_2_/95% O_2_ atmosphere. When the cultures were confluent, microglial cells were isolated by gentle shaking (100 rpm, 1 h) on an orbital agitator at 37 °C [39] and plated on 6-well plates at a density of 42,000 cells/cm^2^. The astrocytes that remained in the flask were isolated by shaking the flasks on an orbital agitator overnight (250 rpm, 18 h). Adhesive cells (mostly astrocytes) were detached by trypsinization, harvested, and plated on hydrophilic surface of 6-well plates, 24-well plates, or 96-well plates at a density of 42,000 cells/cm^2^. All experiments were performed on 5-to-7-day-old secondary cultures and cell purity was checked by staining with antibodies against glial fibrillary acidic protein (GFAP) [40,41].

### 2.4. Cell Treatment

Cultured astrocytes were pre-incubated in the presence or absence of 6-OHDA (120 µM) in serum-free culture medium for 48 h; after that, ODN (10^−10^ M) was administered for an additional 24 h period. When pharmacological substances were used, they were added 30 min before starting incubation with ODN in complete medium. 

### 2.5. Crystal Violet Assay for Viability

Cultured astroglial cells were incubated at 37 °C with fresh serum-free culture medium in the absence or presence of test substances. At the end of the incubation, cells were rinsed three times with PBS (0.1 M, pH 7.4, 37 °C), incubated with formaldehyde (4%, *v*/*v*) for 10 min, washed twice with PBS, and then incubated with crystal violet solution (1%, *w*/*v*) for 10 min. Cells were examined and images were acquired with an Eclipse Ts2 microscope (Nikon, Champigny-sur-Marne, France) equipped with a 3 CCD Sony DXC950 camera interfaced with the Visiolab computerized program (Biocom, Les Ulis, France).

### 2.6. Measurement of Cell Survival

The proportion of surviving cultured cells was quantified by measuring FDA fluorescence. Cultured cells were incubated at 37 °C for 1 h with fresh serum-free culture medium in the absence or presence of 6-OHDA with or without ODN. At the end of the incubation, cells were incubated in the dark with FDA-AM (15 μg/mL, 8 min, 37 °C), rinsed twice with PBS (0.1 M, pH 7.4, 37 °C), and lysed with Tris/HCl solution containing 1% sodium dodecyl sulphate (SDS). Fluorescence intensity (λ excitation = 485 nm and λ emission = 538 nm) was measured using an FL800TBI fluorescence microplate reader (Bio-Tek Instruments, Winooski, VT, USA) or a FlexStation 3 (Molecular Devices, Sunnyvale, CA, USA).

### 2.7. Measurement of Cellular Cytotoxicity

Cultured astrocytes were incubated at 37 °C with fresh serum-free culture medium in the absence or presence of test substances. At the end of the incubation, membrane integrity was assessed as a function of the amount of cytoplasmic LDH released into the medium with an LDH assay kit according to the manufacturer’s instructions. LDH activity was measured at 340 nm with a spectrophotometric microplate reader (Bio-Rad Laboratories, Philadelphia, PA, USA).

### 2.8. Measurement of Intracellular Formation of Reactive Oxygen Species and Nitric Oxide

ROS was detected by measuring the fluorescence of dichlorofluorescein (DCFH) derived from the hydrolysis and oxidation of the non-fluorescent compound DCFH_2_-DA. NO was detected by measuring the fluorescence by the probe DAF-FM DA. Cells seeded into 24-well plates were exposed to 6-OHDA with or without ODN, and at the end of incubation, cells were exposed to fresh medium containing 10 μM cell permeant DCFH_2_DA or DAF-FM DA in serum-free loading medium for 30 min at 37 °C and then washed twice with PBS. Fluorescence was measured with excitation at λ = 485 nm and emission at λ = 538 nm using a microplate reader (Bio-Tek FL800TBI, Instruments, Winooski, VT, USA).

### 2.9. Flow Cytometric Detection of Apoptosis and Necrosis 

Annexin V-FITC/propidium iodide (PI) double staining was used to assess the percentage of apoptotic cells induced by each treatment with the Annexin V/dead cells apoptosis kit (BMS500-FI-300, Invitrogen). Cultured astrocytes seeded into 12-well plates at a density of 0.14 × 10^6^ cells/mL were incubated at 37 °C with fresh serum-free culture medium in the absence or presence of test substances. At the end of the incubation, cells were washed with PBS (0.1 M, pH 7.4), trypsinized, centrifuged (200× *g*, 5 min), and stained with Annexin V-FITC/PI according to the manufacturer’s instructions. Events were analyzed for each sample using the BD FACSCanto II flow cytometer (BD Biosciences, Franklin Lakes, NJ, USA). Data were processed with BD FACSDiva 7 software (BD Biosciences). Four cell subpopulations were evaluated: viable cells (AV^−^/PI^−^), early apoptotic cells (AV^+^/PI^−^), late apoptotic and/or secondary necrotic cells (AV^+^/PI^+^), and necrotic/damaged cells (AV^−^/PI^+^).

### 2.10. Total RNA Extraction and Quantitative RT-PCR

Cultured astrocyte cells seeded into 6-well plates were incubated at 37 °C with fresh serum-free culture medium in the absence or presence of test substances. At the end of treatment, cells were washed with PBS (0.1 M, pH 7.4). Total RNA was extracted and purified by using a NucleoSpin kit (Macherey-Nagel, Hoerd, France). cDNA was synthetized from 1 μg of total RNA with SensiFAST kit (cDNA Synthesis Kit). PCR amplifications were performed with an ABI PRISM 7500 Sequence Detection System (Applied Biosystems, Courtaboeuf, France). Quantitative RT-PCR was performed on cDNA with OZYME kit (One Green FAST qPCR Premix) forward and reverse primers (Table 1). The relative amount of cDNA in each sample was calculated using the comparative cycle threshold (Ct) method, in which the number of targets is given by 2^−ΔΔCt^ using GAPDH as an internal control.

MicroRNA was extracted and purified by using the mirVana™ kit (Thermo Fisher, Waltham, MA, USA) according to the manufacturer’s protocol. Total RNA amount was quantified with the Nanodrop™ One Spectrophotometer and with Qubit™ 4 Fluorometer. Reverse transcription and quantitative PCR was performed using the TaqMan^®^ (MicroRNA Reverse Transcription Kit) and TaqMan^®^ (Fast Advanced Master Mix) from Thermo Fisher.

TaqMan specific probes was used for miR-34b-5p, miR-29a-5p, and miR-21-3p (002617, 002447, and 002493, Thermo Fisher). U6 snRNA (001973, Thermo Fisher) was used as an internal control for normalization, and quantitative RT-PCR was performed on QuantStudio™ 5 (TaqMan, Applied Biosystems, Foster City, CA, USA). Relative quantification of gene expression was based on the comparative cycle threshold (Ct) method, in which the number of targets is given by 2^−ΔΔCt^.

### 2.11. MicroRNA Transfection and 6-OHDA Treatment

Astrocyte transient transfections were performed using the Lipofectamine™ RNAiMAX Transfection Reagent (Invitrogen) under the manufacturer’s instructions. Cells seeded in 6-well plates (0.3 × 10^6^) were transfected for 24 h with 25 nM miR-21-3p inhibitor (MH13039, Stem Loop Sequence: 5′-GACAGCCCAUCGACUGCUGUUG-3′, Thermo Fisher) or non-target miR (miR scramble, 4464058, Thermo Fisher). Astrocyte cells were then incubated at 37 °C for 48 h with fresh complete medium in the absence or presence of 6-OHDA (120 µM). At the end of the incubation, cells were washed with PBS (0.1 M, pH 7.4). miRNA levels were evaluated by qPCR.

### 2.12. Western Blot Analysis for Caspase-3 Protein Expression

Whole-protein extracts were mixed with a sample buffer mix containing N-PER™ (Reagent for Neuronal Protein Extraction) and Phosphatase I, II, and III inhibitor cocktails (Thermo Fisher) under the manufacturer’s instructions. The homogenate was centrifuged (10,000× *g*, 4 °C, 10 min) to collect the proteins contained in the supernatant. Proteins were measured by using the Bradford reagent method and normalized. Proteins were finally denatured in 50 mM Tris–HCl (pH 7.5) containing 20% glycerol, 0.7 M 2-mercaptoethanol, 0.004% (*w*/*v*) bromophenol blue, and 3% (*v*/*v*) SDS at 95 °C for 5 min.

Protein samples (25 μg) were subjected to 12% SDS/polyacrylamide gel electrophoresis (SDS-PAGE) and the gel was transferred onto a nitrocellulose membrane (Bio-Rad Laboratories, Hercules, CA, USA). The membrane was first incubated at room temperature for 1 h in a blocking solution containing 5% skim milk in 50 mM Tris-buffered saline solution completed with 0.1% Tween 20 (TBST). The membranes were then incubated with primary antibodies against caspase-3 (1:1000 dilution, Diagomics, Blagnac, France) or glyceraldehyde-3-phosphate dehydrogenase (GAPDH; 1:50,000 dilution; mAB High Dilution, ABclonal, Woburn, MA, USA) overnight at 4 °C. The day after, membranes were washed with TBST and then incubated in TBS containing 5% skim milk with an HRP anti-rabbit secondary antibody for caspase-3 and anti-mouse secondary antibody for GAPDH (1:5000 dilution; Santa Cruz Biotechnology, Dallas, TX, USA) for 1 h at room temperature and then washed three times with TBST. Again, proteins were revealed using a chemiluminescence detection with Calrity™ Western ECL Substrate kit (Bio-Rad Laboratories) and measured with an image analysis system and software, Image Lab 6.0.1 (Bio-Rad Laboratories).

### 2.13. Rat Cytokine Antibody Array

The effect of ODN on 6-OHDA-induced production of cytokine was measured by using a rat Cytokine Antibody Array Kit (Abcam, Cambridge, UK, ab133992). Cells were incubated at 37 °C with fresh serum-free culture medium in the absence or presence of 6-OHDA (120 μM) with or without ODN. At the end of the incubation, 2 mL of culture media from astrocyte cells culture was collected and then centrifuged at 200× *g* for 10 min. The cytokine membranes were blocked with blocking buffer in room temperature for 1 h. The blocking buffer was removed from the membranes and 2 mL of cultured media was subjected to membrane containing array of antibodies and anti-cytokines. The membrane was first incubated at 4 °C for 24 h in a blocking buffer. Following this step, the membrane was washed with washing buffers (I and II) five times and then incubated with 1X biotin-conjugated anti-cytokines overnight at 4 °C. Membranes were washed again with washing buffers (I and II) five times and incubated with 1X HRP-conjugated streptavidin for 2 h at room temperature. Membranes were washed again with washing buffers (I and II) five times. Membranes were transferred to a plastic sheet and 500 μL of equal volumes of detection buffers C and D was added and incubated for 2 min before imaging. Finally, array images were measured with an image analysis system and software, Image Lab (Bio-Rad Laboratories).

### 2.14. Detection of Cytokine Release by ELISA

Sample cell supernatants (100 μL) were evaluated for release of MCP-1, IL-10, and VEGF using Abcam Rat ELISA Kit (ab219045, ab214566, ab100786). Quantification of this protein release was performed according to the manufacturer’s guidelines. Data were quantified at 450 nm using a FlexStation 3 (Molecular Devices, Sunnyvale, CA, USA).

### 2.15. Bioinformatics Analysis

For network visualization and functional analysis, we used miRNet 2.0 (https://www.mirnet.ca/miRNet/). miRNA–target interaction data were downloaded from miRTarBase and KEGG databases to examine miRNA regulatory network and analytics to gain functional insights. 

### 2.16. Statistical Analysis

Data are expressed as the means ± SEMs from three independent experiments. Statistical analysis of the data was performed by using ANOVA followed by Bonferroni’s test. A *p* value of 0.05 or less was considered statistically significant.

## 3. Results

### 3.1. Delayed Administration of ODN Protects Astroglial Cells from 6-OHDA Toxicity

We have previously shown that the incubation of cultured astrocytes with increasing concentrations of 6-OHDA (10–200 µM) provoked a dose-dependent decrease in the proportion of surviving cells in a study [15]. The concentration of 120 µM 6-OHDA, which killed 50% of cells, was used in all subsequent experiments in order to investigate the potential protective effect of the neuropeptide ODN when added 48 h after the 6-OHDA-induced injury. The administration of graded concentrations of ODN (10^−14^ to 10^−8^ M) to the culture medium suppressed the detrimental action of 6-OHDA in astrocyte survival within 24 h (Figure 1Aa). At a subnanomolar concentration (10^−10^ M), ODN significantly prevented 6-OHDA-induced astroglial cell death. The incubation of astrocytes with ODN alone did not affect cell survival regardless of the time or the dose.

In control conditions or incubated with 10^−10^ M ODN, cells exhibited the typical shapes of healthy astrocytes (Figure 1Ab). In contrast, incubation with 6-OHDA (120 µM, 72 h) caused cell shrinkage and the appearance of long thin processes with the loss of the astrocytic network (Figure 1Ab). Strikingly, the administration of ODN 48 h after the onset of 6-OHDA administration prevented the morphological alterations evoked by 6-OHDA after 24 h of treatment and restored cell density (Figure 1Ab). The addition of subnanomolar concentrations of ODN (10^−10^ M) to the culture medium significantly abolished the action of (120 μM) 6-OHDA on the loss of membrane integrity and LDH leakage (Figure 1Ac). As indicated in the Appendix A (Appendix A), the level of LDH release into a medium is significantly reduced in cell cultures co-treated with 6-OHDA and ODN compared to cells treated with 6-OHDA alone. Concurrently, ODN reduced the increase in LDH leakage induced by 6-OHDA (−48%, ^###^ *p* < 0.001 vs. 6-OHDA-treated astrocytes; Appendix A).

To examine whether ODN could block 6-OHDA-induced intracellular ROS accumulation, astrocyte cells were labeled with CM-H_2_DCFDA, a probe that forms the fluorescent DCF compound upon oxidation by ROS. The incubation of cells with 6-OHDA for 72 h induced a 236.5% increase in DCF fluorescence intensity. The addition of low concentrations of ODN (10^−14^ or 10^−10^ M) to the culture media 48 h after 120 μM 6-OHDA incubation significantly counteracted 6-OHDA-induced intracellular ROS up-production (Figure 1B). At higher concentrations (10^−8^ M), the protective effects of ODN against 6-OHDA evoked astrocyte cell death and intracellular ROS (Figure 1B) and NO (Figure 1C) accumulation was declined. 

Pharmacological studies have indicated that the neuroprotective effect of ODN on astrocytes and neurons is mediated by its metabotropic receptor [14,23]. Herein, to verify whether the desensitization of receptors could be involved in the attenuation of the protective effect of ODN at the dose 10^−8^ M, a pharmacological study was conducted by using selective antagonists of ODN receptors. Cultured astrocytes were pre-incubated with flumazenil (1 µM), a specific antagonist of the CBR-GABA-A receptor, for 30 min (Figure 1D) and with barbadin (100 µM), an inhibitor of GPCR internalization, for 30 min (Figure 1E) with medium alone or with 120 µM 6-OHDA, without or with ODN (10^−10^ M and 10^−8^ M). The results show that the pre-incubation of astrocytes with flumazenil did not impair the protective effect of low concentrations of ODN (10^−10^ M), and ODN was found to be protective at doses of 10^−8^ M (Figure 1D). By blocking the activity of the metabotropic receptor with barbadin, the protective effect of the ODN at 10^−10^ M is suppressed without affecting the attenuation of the ODN effect at 10^−8^ M (Figure 1E). 

### 3.2. ODN Exerts Its Protective Action against 6-OHDA Toxicity by Attenuating Apoptotic Signaling Pathway

To further distinguish the features of apoptotic cells from those of necrotic cells, astrocytes were double-stained by annexin V and PI and flow cytometry analysis was performed to assess viable, early-phase apoptotic, late-phase apoptotic, and necrotic cells. As shown in Figure 2A, untreated astrocyte cultures revealed the following results—Q1: 12.4% necrotic (AV^−^, PI^+^), Q2: 7.4% late-phase apoptotic (AV^+^, PI^+^), Q3: 17.1% early-phase apoptotic (AV^+^, PI^−^), and Q4: 63.1% viable events (AV^−^, PI^−^). However, the analysis of cells treated with 6-OHDA (120 µM, 72 h) showed a significant increase in the number of late-phase apoptotic cells (23.5%) and a reduction in the proportion of viable cells (53.3%). In agreement with its trophic action, ODN exhibited an increase in the proportion of viable cells (75.2%) and a concomitant decrease in constitutive apoptosis. 

To better understand the mechanism by which ODN attenuates 6-OHDA-induced cell apoptosis, we examined the effect of ODN on the expression of the anti-apoptotic gene Bcl-2 and of the proapoptotic gene *Bax* by means of real-time PCR. The exposure of astrocytes to 120 µM 6-OHDA for 54 h produced a decrease (−34.0%, **** *p* < 0.0001) in Bcl-2 mRNA levels (Figure 2Ba) and a concomitant increase (+38.0%, **** *p* < 0.0001) in Bax mRNA levels (Figure 2Bb). The addition of ODN (10^−10^ M) to the culture medium within the last 6 h had no effect on Bcl-2 and Bax mRNA levels but totally suppressed the effect of 6-OHDA on Bcl-2 and Bax expressions. Furthermore, ODN also prevented the 6-OHDA-induced stimulation of caspase-3 gene expression (−75.0 ± 2.3%, ^##^
*p* < 0.01) (Figure 2Bc) and cleavage of caspase-3 (−53.0 ± 1.8%, ^##^
*p* < 0.01) (Figure 2Bd).

### 3.3. Effect of ODN on 6-OHDA-Induced Production of Cytokines

Since 6-OHDA provokes an inflammatory response that facilitates neurodegeneration [42], we investigated the effect of ODN on the production of cytokines and chemokines in 6-OHDA-treated astrocytes. For this, we screened the cytokine/chemokine chips as a means of identifying the key proteins and pathways on the neuroinflammation process. 

Protein array analysis was used to detect 34 proteins including pro- and anti-inflammatory cytokines and neurotrophic factors. The protein array analysis identified 13 differentially expressed proteins in 6-OHDA treated astrocytes compared to control cells (Figure 3Aa). The expression levels of pro-inflammatory cytokines such as CINC 2a, which is also called macrophage inflammatory protein 2-α (MIP2-α); cytokine-induced neutrophil chemoattractant 3 (CINC 3); and TNF-α increased in cells treated with 6-OHDA (120 µM, 72 h). However, the levels of the neurotrophin vascular endothelial growth factor (VEGF), monocyte chemoattractant protein-1 (MCP-1), and lipopolysaccharide-induced CXC chemokine (LIX, also called small-inducible cytokine B5) decreased. The analysis of the cluster heat map from cells subjected to ODN treatment revealed 19 differentially expressed proteins and showed that ODN had slightly modified the levels of proteins compared to control cells and attenuated the effect of 6-OHDA in the overproduction of pro-inflammatory factors (Figure 3Ab).

We next quantified, by ELISA, the levels of MCP-1 (Figure 3B), interleukin-10 (IL-10) (Figure 3C), and VEGF (Figure 3D) released from astrocytes. The levels of MCP-1 and VEGF were particularly low after 120 µM 6-OHDA treatment whereas the levels of IL-10 remained unchanged. The addition of ODN (10^−10^ M) to the culture medium 48 h after 6-OHDA pretreatment restored the levels of MCP-1 and VEGF proteins to control levels and increased IL-10 content above control levels.

Since the secretion of pro-inflammatory cytokines/chemokines such as MCP-1 and IL-10 promotes microglial activation and proliferation [43] and considering the important role of microglial cells in the neuroinflammatory response, we examined whether a conditioned medium collected from untreated astrocytes and 6-OHDA-treated astrocytes with or without ODN could modulate microglial activation and polarization. By using RT-PCR, we showed that the treatment of microglial cultured cells for 6 h with a conditioned medium from 120 μM 6-OHDA-treated astrocytes enhanced the expression of CD86 (+54.0%, * *p* < 0.05), a marker of the pro-inflammatory M1 phenotype (Figure 3Ea), and reduced mRNA levels of CD206 (−36.0%, * *p* < 0.05), a marker of the anti-inflammatory M2 phenotype (Figure 3Eb). In contrast, a conditioned medium collected from ODN + 6-OHDA co-treated astrocytes abolished the effect of 6-OHDA on CD86 up-regulation and CD206 down-regulation in microglial cells. 

### 3.4. Bioinformatics Analysis of the microRNAs Implicated in Parkinson’s Disease 

In the quest to identify miRNAs most associated with PD, a bioinformatics analysis performed using the miRNet 2.0 database [44] revealed the dysregulation of 119 miRNAs in the context of PD (Figure 4A). In order to explore the potential therapeutic effect of certain miRNAs against the toxicity of 6-OHDA and the effectiveness of ODN to modulate their expression in order to potentially counter the deleterious effects of 6-OHDA in cultured astrocytes, we selected three miRNAs candidates, i.e., miR-34b-5p, which regulates the expression of 125 different mRNAs, and miR-29a-5p and miR-21-3p, which target 97 and 83 mRNAs, respectively (Figure 4B). According to our bioinformatics analysis, these three miRNAs were found to converge on target mRNAs, as illustrated in the network (Figure 4B). Indeed, the gene transcripts of PETN, AUMECR1L, and SH3GLB1 were under the dual influence of miR-29a-5p and miR-21-3p, those of XIAP and C18orf32 were under the influence of miR-34b-5p and miR-29a-5p, and the mRNA of the *ZNF546* gene was regulated by miR-34b-5p and miR-21-3p (Figure 4B). It is noteworthy that miR-21-3p controls, at least in part, the expression of Fas ligand (FasL) transcripts [45], which are released intensively from astrocytes in cultures exposed to 6-OHDA and weakly in the presence of ODN (Figure 3Ab). Furthermore, miR-34b-5p seems to play a role in the expression levels of Bcl-2, which were themselves regulated by 6-OHDA and ODN (Figure 4B).

### 3.5. ODN Regulates the Expression of miRNAs, and miR-21 Inhibitor Reduces Cell Death against 6-OHDA on Cultured Astrocytes

Herein, we examined the effect of ODN on some miRNA expressions. Because miRNAs, particularly miR-34b and miR-21-3p, are known to be related in the pathogenesis of PD [33,46], we wondered whether these miRNAs were involved in the protective action of ODN against 6-OHDA neurotoxicity in astrocytes. We first examined the expression levels of these miRNAs in 6-OHDA-treated astrocytes. Quantitative RT-PCR results showed that incubation with 120 µM 6-OHDA had induced a significant reduction in miR-34b-5p (−61.0 ± 1.2%, ** *p* < 0.01) (Figure 5Aa) and miR-29a-5p (−49.0 ± 4%, *****p* < 0.0001) (Figure 5Ab) levels associated with an increase in miR-21-3p expression (+118 ± 2.7%, *** *p *< 0.001) (Figure 5Ac). The addition of ODN (10^−10^ M) 48 h after 6-OHDA administration had suppressed the effect of 6-OHDA on miR-34b-5p and miR-29a-5p down-regulation and miR-21-3p over-expression (Figure 5Aa,b,c). 

To determine whether the regulation of miR-21-3p expression plays a crucial role in the protective action of ODN from cell death induced by 6-OHDA, we next examined the effect of miR-21-3p inhibitor on 6-OHDA-induced cell injuries. The transfection of astrocytes with miR-21-3p inhibitor, which had no effect by itself on cell survival, significantly promoted cell viability in 6-OHDA-treated cultured astrocytes (Figure 5Ba). In addition, miR-21-3p inhibitor prevented the decrease in Bcl-2 mRNA levels induced by 6-OHDA and up-regulated *Bcl-2* gene expression above control values (Figure 5Bb). Furthermore, astrocytes transfected with miR-21-3p inhibitor exhibited *caspase-3* gene transcription at a comparable level to that of cells that had not been exposed to 6-OHDA (Figure 5Bc). 

### 3.6. Inhibition of miR-21-3p Reduces Cytokines and Activates Chemotaxis on Cultured Astrocytes

To confirm that miR-21-3p could be involved in the mechanism of action of ODN, the effect of miR-21-3p inhibitor was examined on the 6-OHDA-induced neuroinflammation response. The same protein array as above was used to screen 34 cytokines and chemokines released from cultured astrocytes transfected with miR-21-3p inhibitor exposed or not to 6-OHDA. The analysis of the cluster heat map of different cultured media indicates that transfection with miR-21-inhibitor slightly regulated the basal production of cytokines and chemokines compared to astrocytes transfected with non-targeting miR inhibitor (control) and identified 28 differentially expressed proteins in 6-OHDA-treated astrocytes (Figure 6Aa,b). As shown in Figure 6B,C, transfection with miR-21-3p inhibitor prevented the effect of 6-OHDA in terms of the decrease in VEGF levels and enhanced the production of IL-10, suggesting that the repression of miR-21-3p expression could account for the protective action of ODN against 6-OHDA-provoked cell death. 

## 4. Discussion

Emerging evidence have shown that miRNAs, post-transcriptional regulators of gene expression, play a pivotal role in neuronal protection, plasticity, damage, and development. For instance, miR-223, miR-153, and miR-22 [46,47] have been proposed as biomarkers for PD. Furthermore, it has been demonstrated that the neuropeptide ODN exerts protective effects against 6-OHDA neurotoxicity in neuronal cultures and in Parkinsonian mice, suggesting that it may represent a promising new therapeutic tool for PD [15,16]. In this in vitro study, we found that the addition of ODN 48 h after the administration of 6-OHDA decreased ROS production, the over-expression of pro-inflammatory genes, and apoptosis in cultured astroglial cells. ODN prevented the down-regulation of miR-34b and miR-29a-5p expression and over-expression of miR-21-3p induced by 6-OHDA. Finally, the down-regulation of miR-21-3p significantly decreased ROS and pro-inflammatory cytokine production, increased anti-inflammatory cytokine expression and the survival of 6-OHDA-treated astrocytes, and decreased apoptosis. 

Our results corroborate that 6-OHDA hinders neuronal cell viability [48,49]. Indeed, 6-OHDA treatment disrupted plasma membrane integrity and increased both LDH leakage as well as intracellular levels of ROS and NO in cultured astrocytes. The incubation of cultured astrocytes with subnanomolar concentrations of ODN dose-dependently prevented the deleterious action of 6-OHDA on LDH release, intracellular ROS accumulation and over-production, and cell viability. Although the beneficial effect of ODN against astrocyte cell death had been reported previously [15,50], this was the first demonstration that ODN is still effective when administered 48 h after the onset of 6-OHDA-induced damages, suggesting that the protective activity of ODN could be exploited for the post-injury treatment of brain insults. Consistent with this observation, it has been shown that the administration of ODN at the subacute period after stroke (3 to 7 days) improves functional recovery and motor recovery over the following month in experimental stroke mice [51]. Studies measuring the infarct volume in ODN precursor-deficient mice have demonstrated the major role of endogenous ODN in reducing neuronal damages induced by stroke [51]. Mass spectrometry, radioimmunoassay, and q-RT PCR analyses reveal that under moderate oxidative stress, ODN biosynthesis and release are increased from cultured astrocytes [26]. This induction of endogenous ODN production by astrocytes in response to exposure to oxidative insults is responsible for (i) the stimulation of endogenous antioxidant systems and thus the rapid resorption of ROS, (ii) the prevention of damages to biomacromolecules, and (iii) protection from apoptosis cell death. In contrast, the blockage of ODN precursor expression by DBI siRNA induces morphological alterations with the loss of membrane integrity and the formation of apoptotic bodies and increases the vulnerability of oxidative stress, inducing astrocyte cell death [26]. Furthermore, the deficiency of the ODN precursor, in an in vivo model of PD, increased brain sensitivity to MPTP neurotoxicity [16]. That ODN might directly act on nigrostriatal dopaminergic neurons is actually a matter of speculation. Thus, the protective effect of ODN on dopaminergic neurons in vivo may result from both a direct effect on neurons and an indirect effect through the protection of glial cells and release of neuroprotective compounds from ODN-activated astrocytes. 

ODN exerts its effects by binding to either CBR that is associated with the GABA-A receptor complex [18] or a still-unidentified GPCR [13,41]. Dose-response experiments indicate that ODN was neuroprotective at low concentrations (in the nanomolar range) while, at higher doses, this action was totally abolished. Such a bell-shaped concentration-response curve has already been observed with ODN with regard to cell proliferation [52], neuronal cell protection [14], and antioxidant enzyme activities in cultured astrocytes [13]. As a matter of fact, the concentrations of ODN required to prevent the deleterious effects of 6-OHDA on astrocytes were in the same range as those that activate the GPCR [23]. In fact, the selective CBR antagonist flumazenil did not impair the protective effect of ODN at a low concentration. Therefore, the attenuation of ODN action at high doses could be accounted for by the desensitization of the ODN metabotropic receptor related to the internalization process [23]. In support of this hypothesis, the inhibition of GPCR endocytosis with β-arrestin/β2-adaptin inhibitor barbadin [53,54] totally abrogated the antiapoptotic activity of ODN from 6-OHDA cell damage on astroglial cells, indicating that the neuroprotective effect of ODN is mediated via a GPCR. Given that β-arrestin-mediated endocytosis contributes to cAMP-PKA-MAPK transduction pathway activation by many GPCRs [53], the inhibitory effect of barbadin on β-arrestin receptor/complex clustering most likely contributes to the blockade of ODN metabotropic receptor-mediated PKA/MAPK-ERK1/2 activation in astrocytes, which could lead to the abrogation of the protective effect of ODN following barbadin treatment

It is widely accepted that the over-production of ROS induces cell death by multiple mechanisms including the damaging of mitochondria, leading to the activation of mitochondria-dependent apoptotic pathways [55,56]. It has been previously shown that ODN is able to prevent the 6-OHDA provoked alteration of mitochondrial integrity in cultured astrocytes [15], and it is well known that mitochondrial membrane permeability is under the control of pro- and anti-apoptotic factors that belong to the Bcl-2 family [57]. Herein, we show that 6-OHDA exerted opposite effects on the expression of Bax, a pro-apoptotic member of the Bcl-2 family, and Bcl-2, an anti-apoptotic factor. The observation that ODN stimulated Bcl-2 expression and totally suppressed the increase in Bax expression induced by 6-OHDA indicates that the peptide is able to control the balance between the pro- and anti-apoptotic factors Bax and Bcl-2 even when administered in a delayed manner after damage induction. As a consequence, the addition of ODN 48 h after the onset of 6-OHDA prevented the stimulatory effect of 6-OHDA on caspase-3 expression and activation. The activation of caspases leading to apoptosis can be divided into two phases corresponding to the early apoptosis occurring during the initiation phase of the apoptosis pathway—phase I—and late apoptosis, representing the final stages of apoptosis when cells engage actively in cell death signaling—phase II of apoptosis or necroptosis [58]. In agreement with its trophic action [51,59], ODN increased the number of viable cells and reduced the proportion of cells, in the early apoptosis phase, compared to that of untreated cells. The addition of ODN to 6-OHDA-treated cells induced a significant decrease in the proportion of apoptosis and necroptosis astroglial cells to a level similar to that observed in cells that had not been exposed to 6-OHDA. Altogether, these observations indicate that ODN prevents apoptotic cell death by inhibiting (i) both the over-expression of the pro-apoptotic protein Bax as well as the repression of the anti-apoptotic protein Bcl-2 and (ii) the drop of the mitochondrial membrane potential responsible for the stimulation of caspase-3 activity. 

Inflammation is another key player involved in the neurotoxic action of 6-OHDA that can impede neuronal cell survival [60], and 6-OHDA-induced neurodegeneration is accompanied by microglial activation and neuroinflammation [61]. Accordingly, our analysis of the present study shows that the production of the neuroinflammatory markers TNFα, IL-6, and CINC-2 was enhanced in 6-OHDA-treated astrocytes. One difficulty in gaining a complete view of the mechanisms involved in the protective effect of ODN against 6-OHDA comes from the fact that ODN did not prevent the increase in the levels of these pro-inflammatory mediators while totally abolishing the induction of others such as IL-1β and IL-13 after 6-OHDA treatment. Interestingly, ODN prevented the action of 6-OHDA on the decreases in protein levels of anti-inflammatory mediator IL-10 and the trophic factor VEGF. Astroglial cells have previously been shown to be a source of trophic factors such as nerve growth factor (NGF), VEGF, and brain-derived neurotrophic factor (BDNF) [62,63]. VEGF stimulates neurite outgrowth in different types of neurons and supports their survival [64]. VEGF is also known to protect neurons and to promote the proliferation of both neuronal cell progenitors as well as glial cells [65]. In vivo studies have shown that ODN is able to reduce both the number of GFAP-positive reactive astrocytes as well as the expression of pro-inflammatory genes in the SNpc of MPTP-treated animals [16]. Furthermore, ODN-knockout mice are more sensitive to MPTP-induced inflammation and have exhibited an increase in the number of activated GFAP-positive cells and IL-6 expression levels [16]. These data suggest that the peptide could block cytokine over-production and dampen the inflammation processes through a modulation of astrocytic activation to a protective–reactive astrocytic status. Activated astrocytes produce and release a panel of protective antioxidant and neurotrophic factors that can reduce microglial activation and neuroinflammation [66]. In fact, we observed that the administration of a conditioned culture medium from ODN-treated astrocytes, but not from naive astrocytes, decreased the 6-OHDA-induced expression of CD86 and M1 pro-inflammatory microglial phenotype marker and suppressed the inhibitory effect of 6-OHDA on the expression of the M2 anti-inflammatory microglial phenotype marker CD206 in cultured microglial cells [67,68]. These data suggest that ODN could offer protection via the inhibition of microglial M1 polarization evoked by 6-OHDA. In this context, it has been demonstrated that the blockading of microglial activation by minocycline offers protection against 6-OHDA and MPTP-induced neurodegeneration in Parkinsonian mice [69,70,71]. It is well known that microglial phenotype regulation is dependent on interaction with molecules released by surrounding cells (neurons or astrocytes) through membrane-bound pattern recognition receptors [72]. When microglia are exposed to anti-inflammatory growth factors such as colony-stimulating factor 1 (CSF-1), neurotrophic factors, neurotrophins, or glial cell-derived factors, their phenotype may change from M1 to M2 [67]. Thus, we assume that the significant amounts of IL-10, IL-4, MCP-1, L-selectin, the CNTF, TIMP1, and CINC3 detected on media from ODN-treated astrocytes exposed to 6-OHDA were, at least in part, responsible for the shifting effect of ODN on polarization to an M2 phenotype. 

A previous study showed that 6-OHDA induces apoptotic effects on astrocytes while ODN treatment reduces cell death [15]. In the present study, we further examined the impact of ODN on specific miRNAs. Our findings highlight the regulatory role of ODN on miRNA expression, specifically in the context of apoptotic and inflammatory pathways. ODN treatment restored the dysregulated miRNA involved in apoptosis caused by 6-OHDA. In ODN treatment against 6-OHDA, we showed that miR-34b and miR-29a-5p were up-regulated in comparison with the 6-OHDA group. MiR-34b-5p has been found to target genes involved in the oxidative stress response known to generate ROS expressions such as heme oxygenase (HO-1) and nuclear factor erythroid 2-related factor 2 (NRF2) [73], reinforcing the importance of the antioxidant mechanism by reducing ROS accumulation. MiR-29a-5p can promote neurite outgrowth to repair brain injury [74], increase endothelial cell permeability and BBB dysfunction, and reduce pro-inflammatory action via suppressing NLRP3 expression [75].

On the other hand, in our study, miR-21-3p exhibited an increase in expression after 6-OHDA treatment. This miRNA has been shown to promote the expression of pro-inflammatory cytokines such as TNF-α and IL-6 and to increase apoptosis by targeting Bcl-2 [76,77]. However, ODN treatment was able to counteract this inflammatory response by activating anti-inflammatory cytokines such as IL-10. These results provide insights into the molecular mechanisms underlying the protective effects of ODN and its potential therapeutic interest in pathological conditions involving apoptosis and inflammation. The administration of ODN has the potential to reduce the expression of miR-21-3p, indicating that inhibiting miR-21-3p may influence the expression of Bcl-2, which could potentially counteract the pro-apoptotic impact triggered by 6-OHDA. 

Treatment with miR-21-3p inhibitor reduced cytokine and chemokine release. In particular, the incubation of cells with 6-OHDA induced a total decrease in VEGF levels while the miR-21-3p inhibitor allowed high VEGF levels in cells exposed to 6-OHDA. EGF is recognized for its role in promoting the proliferation and survival of astrocytes [78]. Moreover, the combined treatment of miR-21-3p inhibitor and 6-OHDA resulted in a heightened release of IL-10 compared to treatment with 6-OHDA alone. This suggests that the miR-21-3p inhibitor enhances the anti-inflammatory effect of 6-OHDA by promoting the release of IL-10. In addition, the combination of the miR-21-3p inhibitor and 6-OHDA treatment resulted in an elevated release of the ciliary neurotrophic factor (CNTF). The CNTF can stimulate the secretion of multiple growth factors and cytokines from astrocytes, including BDNF, NGF, and IL-6 [79,80]. These factors play pivotal roles in neuronal survival, growth, and immune responses in the brain. The increase in CNTF release suggests that the miR-21-3p inhibitor, in combination with 6-OHDA, influences astrocytic functions and their ability to release important neurotrophic factors. miR-21-3p has been identified as a regulator of FASL [81], which is a transmembrane protein associated with apoptosis and cell death processes [82]. In our data, we observed a reduction in FASL levels when we transfected the cells with miR-21-3p inhibitor before treatment with 6-OHDA. Collectively, these results highlight the role of miR-21-3p in modulating apoptosis in astrocytes exposed to 6-OHDA.

## 5. Conclusions 

In summary, our results demonstrate that ODN at low doses exerts a protective effect during the late apoptosis phase but also modulates the release of cytokines and chemokines from astrocytes, probably after binding to its GPCR. These findings contribute to a better understanding of the mechanisms involved in neuroinflammatory processes and may have implications for the development of therapeutic strategies targeting microglial polarization in neurological disorders, especially in Parkinson’s disease. ODN plays a role in the regulation of some microRNAs, creating a complex interaction between miR-21-3p, astrocytes, and neuroinflammation, highlighting the therapeutic potential of targeting miR-21-3p in neurodegenerative conditions. Altogether, our data bring additional evidence that miRNA open new perspectives for the gene therapy of neurodegenerative diseases associated with oxidative stress and inflammation.

## Figures and Tables

**Figure 1 cells-13-01188-f001:**
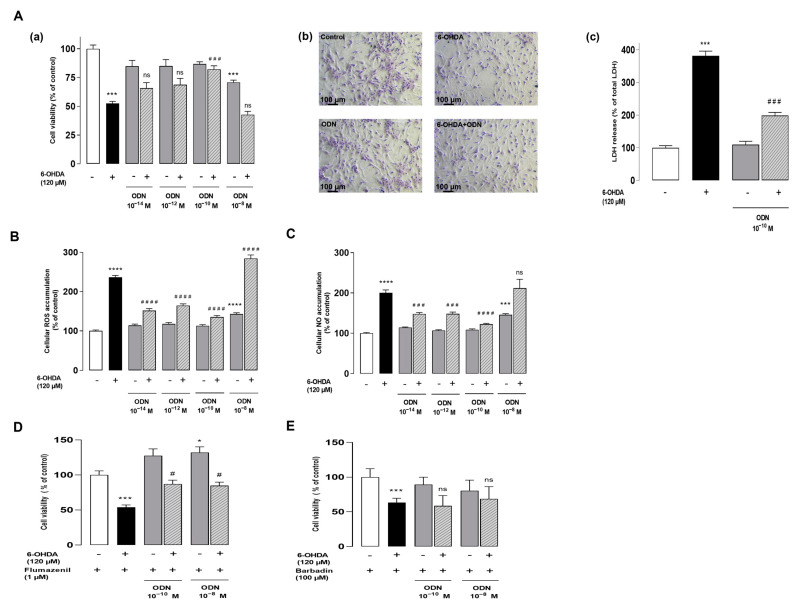
ODN protects cultured astroglial cells from the toxicity induced by 6-OHDA. (**Aa**) Effect of increasing doses of ODN (10^−14^ to 10^−8^ M) on the survival of cultured astrocytes after 72 h of incubation in the presence or absence of 6-OHDA (120 µM); the ODN was administered during the last 24 h. (**Ab**) Phase-contrast images illustrating the protective effect of ODN (10^−10^ M) on morphological changes in astrocytes. (**Ac**) Effect of ODN (10^−10^ M) on the release of lactate dehydrogenase. (**B**,**C**) Effect of ODN on 6-OHDA-induced intracellular ROS and NO accumulation. (**D**,**E**) Cells were pre-incubated with flumazenil (1 µM) or barbadin (100 µM) for 30 min with medium alone or with 120 µM 6-OHDA, without or with ODN (10^−10^ M and 10^−8^ M). Cell survival was quantified by measuring FDA fluorescence intensity; the results are expressed as percentages of control. Each value is the mean (±SEM) of at least 12 different wells from 3 independent experiments. We used ANOVA followed by Bonferroni’s test: *** *p* < 0.001; **** *p* < 0.0001 vs. control. ^#^ *p* < 0.05; ^###^ *p* < 0.001; ^####^ *p* < 0.0001 vs. 6-OHDA-treated cells; ns: not statistically different vs. 6-OHDA-treated cells.

**Figure 2 cells-13-01188-f002:**
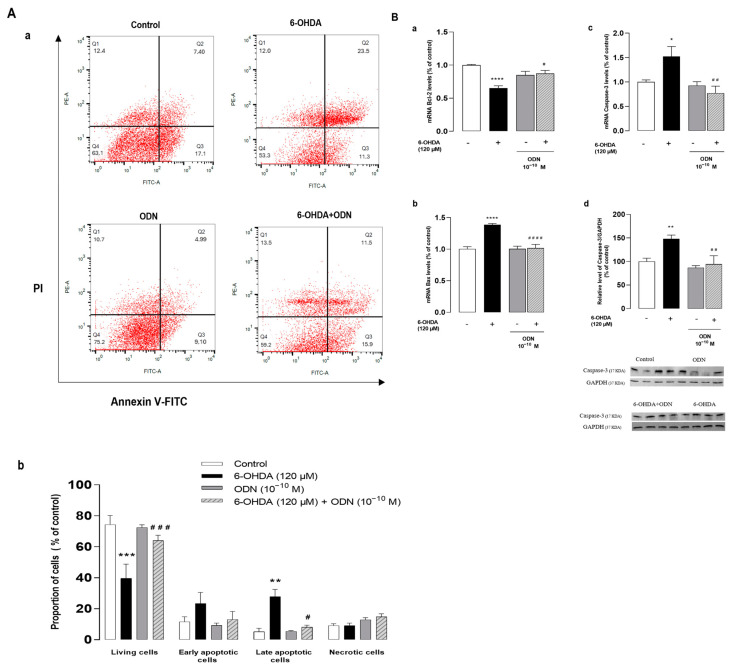
Effects of ODN on the expression of pro- and anti-apoptotic genes and on the level of cleaved caspase. (**Aa**,**b**). Cultured astrocytes were incubated for 72 h in the presence or absence of 6-OHDA (120 µM) and ODN (10^−10^ M) was administered during the last 24 h. The cells were double-stained with annexin V (FITC-A, AV) and propidium iodide (PE-A, PI) and divided into 4 quartiles: living cells (Q4, AV^−^/PI^−^), early apoptotic cells (Q3, AV^+^/PI^−^), late apoptotic cells (Q2, AV^+^/PI^+^), and necrotic cells (Q1, AV^−^/PI^+^). (**Ba**,**b**,**c**) Cultured astrocytes were incubated for 54 h in the presence or absence of 6-OHDA (120 µM) and ODN (10^−10^ M) was administered during the last 6 h. mRNA levels of Bax, Bcl-2, and caspase-3 were quantified by RT-PCR and normalized to GAPDH expression used as an internal control. (**B****d**) Densitometric analysis of cleaved caspase-3 protein levels in astrocytic cells cultured for 72 h in the presence or absence of 6-OHDA (120 µM). ODN (10^−10^ M) was administered during the last 24 h. Photographs illustrate caspase-3 expression after immunoblotting and graphs display the relative abundance of proteins measured by densitometry of the bands obtained in immunoblots and standardized with GAPDH. Each value is the mean (±SEM) of at least 6 different wells from 3 independent experiments. We used ANOVA followed by Bonferroni’s test: * *p* < 0.05; ** *p* < 0.01; *** *p* < 0.001; **** *p* < 0.0001 vs. control. ^#^ *p* < 0.05; ^##^ *p* < 0.01; ^###^ *p* < 0.001; ^####^ *p* < 0.0001 vs. 6-OHDA-treated cells.

**Figure 3 cells-13-01188-f003:**
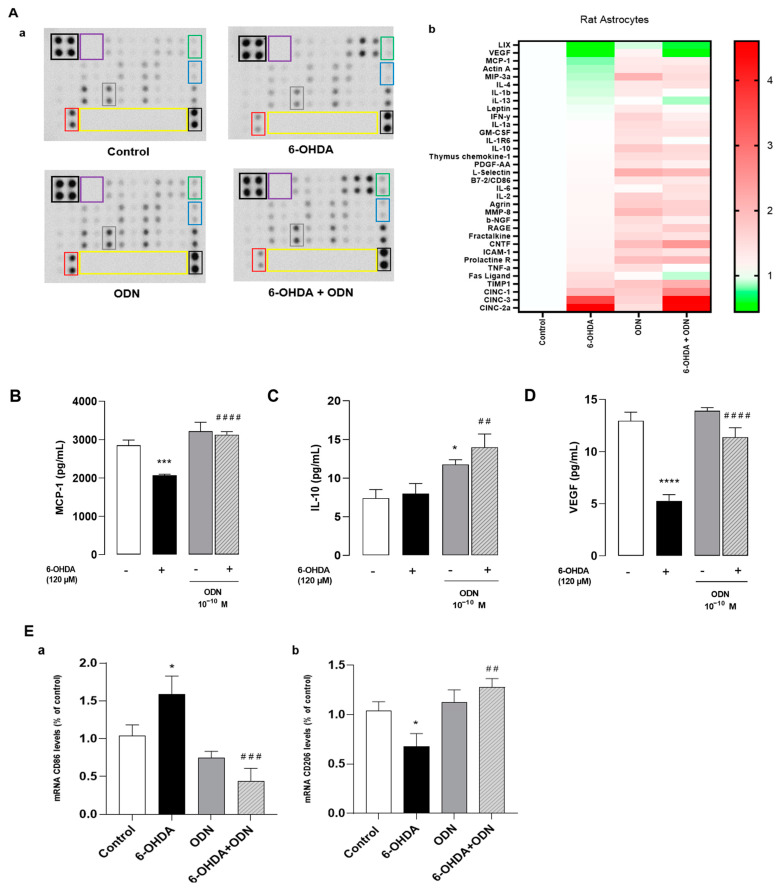
Effects of ODN on the release of cytokines and chemokines on cultured astrocytes. (**A**) Cultured astrocytes were incubated for 72 h in the presence or absence of 6-OHDA (120 µM) and ODN (10^−10^ M) was administered during the last 24 h. (**Aa**) Representative images of the levels of release of 34 proteins in the culture supernatant. The outlined areas represent technical replicates of MCP-1 (gray), IL-10 (blue), VEGF (red), CNTF (green), positive control (black), negative control (purple), and empty well (yellow). (**Ab**) Heat map representation. The secretions of MCP-1 (**B**), IL-10 (**C**), and VEGF (**D**) were quantified using ELISA kits. (**E**) Microglial cells were treated for 6 h with the supernatant from cultured astrocytes previously incubated for 72 h in the presence or absence of 6-OHDA (120 µM) and ODN (10^−10^ M) during the last 24 h. (**Ea**,**b**) After 6 h, mRNA levels of CD86 and CD206 were quantified by RT-PCR and normalized to GAPDH expression used as an internal control. Each value is the mean (±SEM) of at least 6 different wells from 2 independent experiments. We used ANOVA followed by Bonferroni’s test: * *p* < 0.05; *** *p* < 0.001; **** *p* < 0.0001 vs. control. ^##^ *p* < 0.01; ^###^ *p* < 0.001; ^####^ *p* < 0.0001 vs. 6-OHDA-treated cells.

**Figure 4 cells-13-01188-f004:**
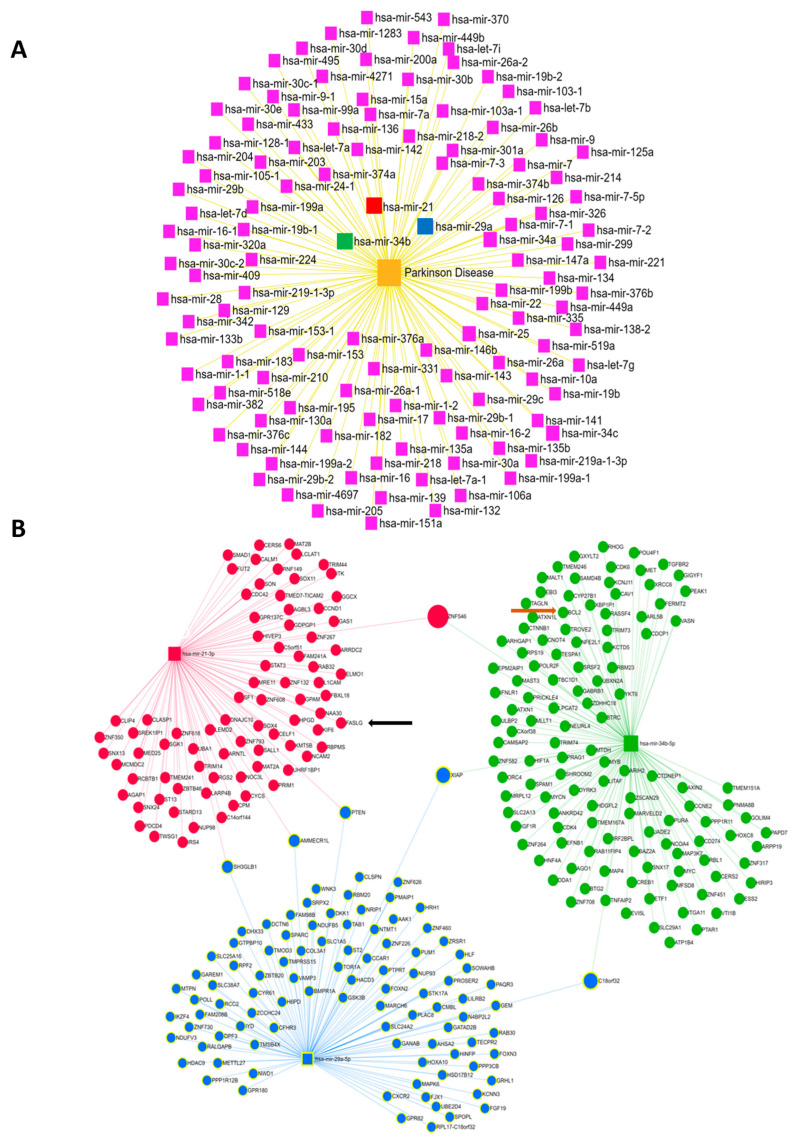
Bioinformatics analysis of the miRNAs most implicated in PD. (**A**) Diagram illustrating the dysregulated expression of miRNAs in patients with PD. (**B**) Diagrams illustrating the interactions between the 3 selected miRNAs, MiR-21-3p (red), miR-34b-5p (green), and miR-29a-5p (blue), and their target mRNAs in humans. The interactions between functional networks are positioned at the interfaces of the three networks for each selected miRNA. Shown: FASL mRNA (black arrow), Bcl2 mRNA (brown arrow).

**Figure 5 cells-13-01188-f005:**
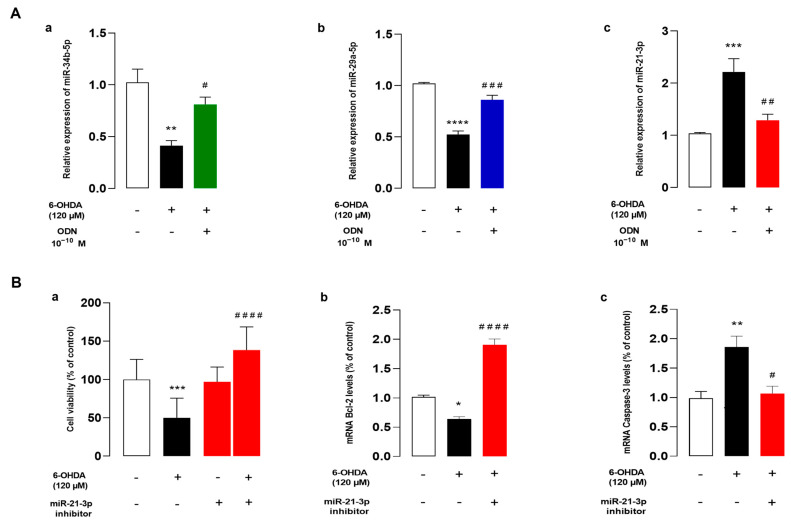
ODN regulates the expression of miRNAs and miR-21 inhibitor reduces cell death against 6-OHDA on cultured astrocytes. (**Aa**,**b**,**c**) Cultured astrocytes were pre-incubated for 48 h in the absence or presence of medium alone or with 120 µM 6-OHDA and then incubated for 6 h with ODN (10^−10^ M) alone or with 120 µM 6-OHDA. Extraction of miRNA was performed and miR-34b-5p, miR-29a-5p, and miR-21-3p mRNA levels were quantified by RT-PCR. Data were corrected using the U6 signal as an internal control and the results are expressed as percentages of controls. Each value is the mean (±SEM) of at three different wells from independent experiments. We used ANOVA followed by Bonferroni’s test: * *p* < 0.05; ** *p* < 0.01; *** *p* < 0.001; **** *p* < 0.0001; ^#^ *p* < 0.05; ^##^ *p* < 0.01; ^####^ *p* < 0.0001. (**B**) Cultured astrocytes were transfected with scramble miRNA alone or with miR-21-3p inhibitor (25 nM). (**Ba**) Cell survival was quantified by measuring FDA fluorescence intensity and the results were expressed as percentages of control. (**Bb**,**c**) Bcl-2 and caspase-3 mRNA levels were quantified by RT-PCR. Data were corrected using the GAPDH signal as an internal control and the results were expressed as percentages of controls. Each value is the mean (±SEM) from at least 12 different wells from 3 independent cultures. We used ANOVA followed by Bonferroni’s test: * *p* < 0.05; ** *p* < 0.01; *** *p* < 0.001; **** *p* < 0.0001 vs. Control. ^#^ *p* < 0.05; ^##^ *p* < 0.01; ^###^ *p* < 0.001; ^####^ *p* < 0.0001 vs. 6-OHDA-treated cells.

**Figure 6 cells-13-01188-f006:**
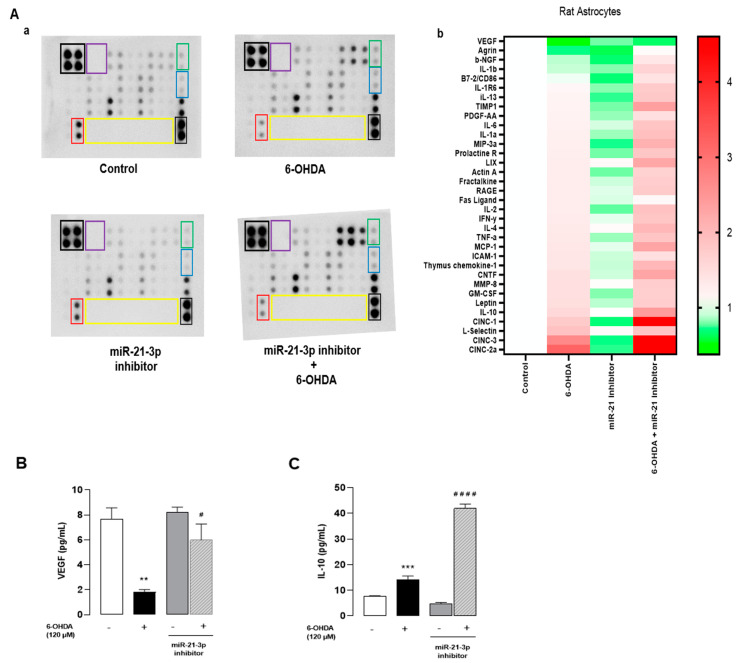
Inhibition of miR-21-3p reduces cytokines and activates chemotaxis on cultured astrocytes. Cultured astrocytes were transfected for 24 h with either an miR-21-3p inhibitor or a control miRNA (scramble), then treated in the presence or absence of 6-OHDA (120 µM) for 48 h. (**Aa**) Representative images of the levels of release of 34 proteins in the culture supernatant. The outlined areas represent technical replicates of VEGF (red), IL-10 (blue), CNTF (green), positive control (black), negative control (purple), and empty well (yellow). (**Ab**) Heat map representation. (**B**,**C**) Secretion of VEGF and IL-10 was performed by Elisa kit. Each value is the mean (±SEM) from at least 12 different wells from 3 independent cultures. We used ANOVA followed by Bonferroni’s test: ** *p* < 0.01; *** *p* < 0.001 vs. Control. ^#^ *p* < 0.05; ^####^ *p* < 0.0001 vs. 6-OHDA-treated cells.

**Table 1 cells-13-01188-t001:** Sequences of the primers used for real-time PCR experiments.

Gene	GenBank Accession Number	Sequence
*Caspase-3*	NM_012922.2	
	Forward	5-AATTCAAGGGACGGGTCATG-3
	Reverse	5-GCTTGTGCCGTACAGTTTC-3
*Bax*	NM_017059.2	
	Forward	5-TGCAGAGGATGATTGCTGATGT-3
	Reverse	5-CAGCTGCCACACGGAAGAA-3
*Bcl-2*	NM_016993.2	
	Forward	5-GGCTGGGATGCCTTTGTG-3
	Reverse	5-CAGCCAGGAGAAATCAAACAGA-3
*CD 86*	NM_020081.2	
	Forward	5-GTGGAAAGGGGCTGTTGATTGG-3
	Reverse	5-TTCTGCCTCTCAGCCAGTTACC-3
*CD 206*	NM_001106123.2	
	Forward	5-GAACGAGAGGTCACAGAGCAGT
	Reverse	5-TACCCCTCACATCTCCCTCACA
*GAPDH*	NM_017008.4	
	Forward	5-CAGCCTCGTCTCATAGACAAGATG-3
	Reverse	5-CAATGTCCAACTTTGTCACAAGAGAAA-3

## Data Availability

All data are available at the Laboratory of Neuroendocrine, Endocrine and Germinal Differentiation and Communication (NorDiC), F-76000 Rouen, France and at LR18ES03 Laboratory of Neurophysiology, Cellular Physiopathology and Valorisation of Biomol-ecules, 2092 Tunis, Tunisia.

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
