# Peer review of "Octadecaneuropeptide, ODN, Promotes Cell Survival against 6-OHDA-Induced Oxidative Stress and Apoptosis by Modulating the Expression of miR-34b, miR-29a, and miR-21in Cultured Astrocytes"

_cells, 2024, doi:10.3390/cells13141188_

Round 1

Reviewer 1 Report

Comments and Suggestions for Authors

Bourzam et al. present an intriguing study on the role of ODN in counteracting 6-OHDA-induced toxicity in astrocytes. However, before considering this article for publication, the following questions and issues need to be addressed:

1. Title: The title appears to be a copy-paste from the first sentence of the abstract. It should be clarified and made more specific.

2. Abstract: The sentence "6-OHDA-treated astrocytes exhibited overexpression of miR-34b (-61%), miR-29a (-49%), and miR-21 (118%) associated with a knockdown of miR-21" is unclear. The use of negative percentages for overexpression is confusing. If it indicates overexpression, the minus sign should be clarified or corrected.

3. Animals: The sex and age of the rats used in the experiment are not specified. Additionally, the rationale for selecting the specific age/sex should be provided.

4. Focus on Astrocytes: The authors should explain why they are focusing on astrocytes and not investigating the role of ODN on 6-OHDA toxicity in dopamine neurons, which are the primary cell population affected in Parkinson's Disease (PD).

5. Figures and Images: The readability of the figures and images is very poor. For the Western Blot image, the authors must provide the entire membrane, as it currently looks like cuts from different blots, which is not acceptable.

Comments on the Quality of English Language

Minor editing

Author Response

Point-by-point answers to the reviewers’ comments

Reviewer # 1:

Bourzam et al. present an intriguing study on the role of ODN in counteracting 6-OHDA-induced toxicity in astrocytes. However, before considering this article for publication, the following questions and issues need to be addressed:

Authors Comment:

We thank the reviewer for his/her constructive and encouraging comment.

  1. Title: The title appears to be a copy-paste from the first sentence of the abstract. It should be clarified and made more specific.

Author Response:

We had attributed a title to the manuscript and it seems that a mistake was made when we submitted the article in the title section. We apologize for this error and here is the title “Octadecaneuropeptide, ODN, promotes cell survival against 6-OHDA-induced oxidative stress and apoptosis by modulating the expression of miR-34b, miR-29a and miR-21 in cultured astrocytes that is now mentioned in the revised version of the manuscript.

  1. Abstract: The sentence "6-OHDA-treated astrocytes exhibited overexpression of miR-34b (-61%), miR-29a (-49%), and miR-21 (118%) associated with a knockdown of miR-21" is unclear. The use of negative percentages for overexpression is confusing. If it indicates overexpression, the minus sign should be clarified or corrected.

Author Response:

We agree with the referee that the sentence is confused in its meaning and was rephrased to provide the correct explanation as follows: "6-OHDA-treated astrocytes exhibited overexpression of miR-21 (+118%) associated with a knockdown of miR-34b (-61%) and miR-29a (-49%). Thus, this is now indicated in abstract section of the revised version lines 31 to 33).

  1. Animals: The sex and age of the rats used in the experiment are not specified. Additionally, the rationale for selecting the specific age/sex should be provided.

Author Response:

We thank reviewer for pointing this out,  astrocyte cell culture are isolated from neonatal cortex of both sex, and this has been clarified in the Material and Methods section page 3, lines 31 to 32 “Secondary cultures of rat cortical astrocytes were prepared from 1- or 2-day-old Wistar rats of both sexes as previously described (Zarei‑Kheirabadi e al., 2020 Molecular Biology Reports https://doi.org/10.1007/s11033-020-05272-2)”.

To be consistent with our previous published studies, we chose newborn rats from both sexes. However, we are aware that some sex-specific pathways of cell death and survival following injuries can arise as shown by Mingyue et coll. (2008 J Neurosci Methods 171: 214-217) and Zhang et coll. (2002 Molecular Brain Research 103: 1-11).

  1. Focus on Astrocytes: The authors should explain why they are focusing on astrocytes and not investigating the role of ODN on 6-OHDA toxicity in dopamine neurons, which are the primary cell population affected in Parkinson's Disease (PD).

Author Response:

As dopaminergic neurons are the target neurons in Parkinson's disease, we have already shown that i.c.v. administration of ODN exerts a potent neuroprotective effect against MPTP-induced degeneration of nigrostriatal DA neurons in mice. Currently, the involvement of astrocytes in the pathophysiology of neurological disorders, including Parkinson neurodegenerative disease is highlighted (Zhou et al., 2019 CNS Neurosci Ther  6:665-673. ). Indeed, it has been reported that injection of 6-OHDA within the nigrostriatal pathway produces a loss of astrocytes associated with the degeneration of dopaminergic neurons (Espinosa-Oliva et al. 2014 Neurotoxicology 41:89-101.). As loss of glial cells may strongly affect neuronal survival, protection of astrocytes from oxidative insults appears beneficial to neuronal survival and essential to maintain brain functions (Hart and Karimi-Abdolrezaee 2021 J. Neurosci. Res. 10:2427-246). We hypothesize that by promoting the tolerance of astrocytes against oxidative stress, ODN may increase the proportion of healthy astrocytes and thus may act to promote neuron survival. Alternatively, ODN may also indirectly act on dopaminergic neurons through activation of glial cells and release of neuroprotective factors. This point has now been discussed page 19 lines 35 to 40 as follows:

“That ODN might directly act on nigrostriatal DA neurons is actually a matter of speculation. Thus, the protective effect of ODN on DA neurons in vivo may result of both a direct effect on neurons and an indirect effect, through the protection of glial cells and release of neuroprotective compounds from ODN-activated astrocytes”.

  1. Figures and Images: The readability of the figures and images is very poor. For the Western Blot image, the authors must provide the entire membrane, as it currently looks like cuts from different blots, which is not acceptable.

Author Response:

The entire Western-blot membranes are provided and the quality of images and figures has been improved.

Reviewer 2 Report

Comments and Suggestions for Authors

The work by Bourzam and colleagues explores mechanisms by which octadecaneuropeptide (ODN) can protect astrocytes from damage due to 6-OHDA.  The work is fairly extensive, interesting, and has important implications.

Specific comments.

1) a title to the paper is needed, somehow the first two sentences of the abstract were used instead.

2) In abstract, line 10: It is potentially confusing to state that "6-OHDA-treated astrocytes  exhibited overexpression of miR-34b (- 61%)..."  It reduced miR-34b by 61%.  Same for miR-29a (-49%).

3) Many of the citations are incorrect.  The authors should go through the manuscript and make sure that all the citations align correctly with the statements made.  For example, reference #4 where first cited refers to 2 patients in trials, not the summary statement about neuroprotection.  Another example: at the bottom of page 3, citations 34-37 are noted as referring to miRNAs and PD, but  they do not appear to be on that topic.  Sometimes the refs are just off by 1 (e.g. reference "#43" on page 12 should be #42).

4) Section 2.2 Chemicals, is it possible to be more specific in defining "antibiotic-antimycotic solution"?  Maybe a product number if nothing else.

5) Can the authors comment on the need to use 120 µM 6-OHDA while low µM 6-OHDA is toxic to neurons.

6) Please estimate the % neurons, % astrocytes in their astrocyte cultures.  Could neuronal death be contributing to LDH release (and other markers) in their study?

7) page 9, 1st full paragraph. The text refers to a -182% change in LDH leakage.  This value doesn't give the reader an idea of the magnitude of the effect (which appears to be about a 60% decrease in 6-OHDA stimulated LDH release). It might be more meaningful to describe what % of the signal is decreased by ODN.  That is, take the + 6-OHDA minus no 6-OHDA as 100% as the level of 6-OHDA stimulation of LDH release and calculate what % decrease occurs by ODN co-incubation.

7) Figure 1 legend.  Unusual to report that some values are ± SEM or are ± SD.  Can the authors indicate which experiments have which?

8) Figure 2 legend.  "Each value is the mean (±SEM) of at least 4 different wells from independent experiments".  Can the authors indicate how many wells from how many idependent experiments?  For example, is the 3 wells from one experiment and 1 well from another experiment?  Or 1 well each from 4 independent experiments?

Same question for other experiments, e.g. Figures 5, 6.

9) Bottom of page 12.  In the conditioned medium experiments, wasn't 6-OHDA (or its oxidative products) still present in the conditioned media?

10) typos: P 16,  line 10, 6-OHD should be 6-OHDA.  Page 2, line 1, "targets" should be "target".

11) Discussion, bottom of page 19. The discussion about the potential role of internalization of the ODN receptor was confusing to me.  It sounded like high concentrations of ODN are expected to cause desensitization by causing internalization and that was why high concentrations were ineffective.  But the internalization inhibitor blocked ODN activity.  Could it be that the internalization inhibitor blocked beta arresting/ beta1-adapting mediated activity of the ODN receptor?

12) The ability of ODN to prevent damage and signaling when given 48 hours after 6-OHDA is remarkable.  Can the authors mention the normal timeline of the 6-OHDA-induced damage in this system and when different signals are getting turned on/off after 6-OHDA addition?

Comments on the Quality of English Language

Mostly OK.  The discussion paragraphs could be smaller with more distinct topic sentences - it helps with the readability.

Author Response

Point-by-point answers to the reviewers’ comments

Reviewer # 2

The work by Bourzam and colleagues explores mechanisms by which octadecaneuropeptide (ODN) can protect astrocytes from damage due to 6-OHDA.  The work is fairly extensive, interesting, and has important implications.

Authors Comment:

We thank the reviewer for his/her constructive and encouraging comment.

Specific comments.

1) a title to the paper is needed, somehow the first two sentences of the abstract were used instead.

Author Response

We had attributed a title to the manuscript and it seems that a mistake was made when we submitted the article in the title section. We apologize for this error and here is the title “Octadecaneuropeptide, ODN, promotes cell survival against 6-OHDA-induced oxidative stress and apoptosis by modulating the expression of miR-34b, miR-29a and miR-21 in cultured astrocytes that is now mentioned in the revised version of the manuscript.

2) In abstract, line 10: It is potentially confusing to state that "6-OHDA-treated astrocytes exhibited overexpression of miR-34b (- 61%)..."  It reduced miR-34b by 61%.  Same for miR-29a (-49%).

Author Response

We agree with the referee that the sentence is confused in its meaning and was rephrased to provide the correct explanation as follows: "6-OHDA-treated astrocytes exhibited overexpression of miR-21 (+118%) associated with a knockdown of miR-34b (-61%) and miR-29a (-49%). Thus, this is now indicated in abstract section of the revised version.

3) Many of the citations are incorrect.  The authors should go through the manuscript and make sure that all the citations align correctly with the statements made.  For example, reference #4 where first cited refers to 2 patients in trials, not the summary statement about neuroprotection.  Another example: at the bottom of page 3, citations 34-37 are noted as referring to miRNAs and PD, but  they do not appear to be on that topic.  Sometimes the refs are just off by 1 (e.g. reference "#43" on page 12 should be #42).

Author Response

The manuscript has been carefully edited and all references have been hopefully cited.

 4) Section 2.2 Chemicals, is it possible to be more specific in defining "antibiotic-antimycotic solution"?  Maybe a product number if nothing else.

Author Response:

The reference of the antibiotic-antimycotic solution used in cell cultures is now specified in the section 2.2 Chemicals page 4 of the revised version of the manuscript.

5) Can the authors comment on the need to use 120 µM 6-OHDA while low µM 6-OHDA is toxic to neurons.

Author Response

We have previously shown that neurons are sensitive to 30 µM 6-OHDA toxicity, a dose 4 times lower than those used for astrocytes (120 µM). We observed that exposure of cultured astrocytes to 6-OHDA concentrations above 30 μM reduced astrocyte viability in a concentration, and time-dependent manner. The resistance of astrocytes to lower doses of the toxin can be ascribed to their lower sensitivity to 6-OHDA toxicity when compared to neurons (Kaddour et al. 2013 J Neurochem 4:620-33; Datta et al., 2018 Mol Neurobiol 7:5505-5525). This can be explained by the higher antioxidant capacity of astrocytes (Baxter and  Hardingham 2016 Free Radic Biol Med 100:147-152)

6) Please estimate the % neurons, % astrocytes in their astrocyte cultures.  Could neuronal death be contributing to LDH release (and other markers) in their study?

Author Response

We have previously controlled the cell purity of secondary culture astrocytes by immunofluorescence using GFAP (astrocytes labeling), rabbit anti-CD68 (microglia labeling) and sheep anti-olig2 (oligodendrocyte labeling), and we have observed that in our culture condition, more than 98% of the cells were labeled with GFAP antibodies (Douiri et al., 2017 Journal of Neurochemistry 137 : 913–930 ).

We therefore assume that neuron population is completely absent in our astrocyte cell culture and that there is no interference from LDH released by neurons in LDH activity measurement.

7) page 9, 1st full paragraph. The text refers to a -182% change in LDH leakage.  This value doesn't give the reader an idea of the magnitude of the effect (which appears to be about a 60% decrease in 6-OHDA stimulated LDH release). It might be more meaningful to describe what % of the signal is decreased by ODN.  That is, take the + 6-OHDA minus no 6-OHDA as 100% as the level of 6-OHDA stimulation of LDH release and calculate what % decrease occurs by ODN co-incubation.

Author Response

We agree with the reviewer that it would be interesting to examine the magnitude of LDH leakage in 6-OHDA plus ODN-cotreated cells versus 6-OHDA-treated cultured astrocytes. Thus, we have now included some data on LDH release, which indicate that 6-OHDA-induced increase in LDH leakage is reduced by 48% by the administration of ODN.

The new data, shown in supplementary data figure S1, indicate that the level of LDH release into medium is significantly reduced in cell cultures co-treated with 6-OHDA and ODN compared to cells treated with 6-OHDA alone. Concurrently, ODN reduced the increase of LDH leakage induced by 6-OHDA (-48% vs 6-OHDA-treated astrocytes; Fig S1). These data have been indicated in the new supplementary data figure S1 and in the result section page 9 lines 13 to 19.

7) Figure 1 legend.  Unusual to report that some values are ± SEM or are ± SD.  Can the authors indicate which experiments have which?

Author Response

Depending the number of experiments and /or replicates we used either values ± SEM. Data are now explained.

8) Figure 2 legend.  "Each value is the mean (±SEM) of at least 4 different wells from independent experiments".  Can the authors indicate how many wells from how many idependent experiments?  For example, is the 3 wells from one experiment and 1 well from another experiment?  Or 1 well each from 4 independent experiments?

Same question for other experiments, e.g. Figures 5, 6.

Author Response

Each value is the mean (± SEM) from at least 12 different wells from 3 independent cultures. This has been clarified and as recommended, the number of wells for each analysis is now provided in the legend of all figures.

9) Bottom of page 12.  In the conditioned medium experiments, wasn't 6-OHDA (or its oxidative products) still present in the conditioned media?

Author Response

We agree with the reviewer that it would be interesting to detect 6-OHDA and/or its oxidative products in the medium, since extracellular autooxidation of 6-OHDA produces a large array of ROS which are likely responsible for part of the toxicity of the molecule (Raicevic et al. 2005 Ann N Y Acad Sci 1048:400-405.). Previous experiments that we have conducted indicate that exposure of cultured astrocytes to 6-OHDA (120 μM; 72h) provoked a significant increase of intracellular level of ROS  i.e. H2O2 and O2° (Kaddour et al., 2019 Journal of Molecular Neuroscience). These data are in accordance with the expression of  catecholamine transporters by astroglila cell which can rapidly promote cytosol accumulation of 6-OHDA and generation of ROS after deamination of the toxin.  Despite these observations, it will be interesting to carry out further experiments to measure the extracellular level of 6-OHDA and ROS in the culture supernatant of 6-OHDA-treated astrocyte

10) typos: P 16,  line 10, 6-OHD should be 6-OHDA.  Page 2, line 1, "targets" should be "target".

Author Response

We apologize for these errors, which have been corrected as highlighted in yellow. Furthermore, the entire text has been checked carefully to remove other typo errors.

11) Discussion, bottom of page 19. The discussion about the potential role of internalization of the ODN receptor was confusing to me.  It sounded like high concentrations of ODN are expected to cause desensitization by causing internalization and that was why high concentrations were ineffective.  But the internalization inhibitor blocked ODN activity.  Could it be that the internalization inhibitor blocked beta arresting/ beta1-adapting mediated activity of the ODN receptor?

Author Response

Previous studies indicate that ODN protects glial cells from oxidative stress through activation of the metabotropic ODN receptor and adenylate cyclase/cAMP/PKA/MAPK/ERK-dependent mechanism. Interestingly, it has been shown that Barbadin, which prevents trafficking of receptor/β-arrestin complexes to intracellular compartments clathrin-coated pits (CCPs) and endosomes, completely inhibits vasopressin receptor type 2 (V2R)-stimulated ERK1/2 activation and blunts both  β2-adrenergic and V2R receptors-promoted cAMP accumulation, supporting the concept of β-arrestin/AP2-dependent signaling for G protein-dependent (Beautrait et al., 2017 Nature Communications DOI: 10.1038/ncomms15054.).  These results are therefore consistent with the notion that GPCR-stimulated intracellular transduction pathway occurs following clustering of receptor/ β-arrestin in CCPs and endocytosis of the complex in endosomes. We therefore hypothesize, as has been recently shown for some GPCR receptors i.e. Bradykinin B2 receptor and β-adrenergic receptor (Zimmerman et al., 2011 Cell  Signal. 23: 648–659; Eichel et al.,  2016 Nat. Cell Biol. 18: 303–310), that ODN metabotoropic receptor-stimulated PKA/MAPK/ERK activation could occur after receptor/β-arrestin clustering in CCPs or endosomes. Thus, the inhibitory effect of Barbadin for clustering receptor/ β-arrestin complexes most likely contributes to the blockade of ODN metabotropic receptor-mediated activation of PKA/MAPK-ERK1/2 in astrocytes and thus leading to the inhibition of the protective effect of ODN following Barbadin treatment.  

To gain insight on the inhibitory properties of Barbadin on ODN metabotropic receptor activity, we will study in future experiment the effect of Barbadin on ODN-induced cAMP formation and/or ERK1/2 phosphorylation in cultured astrocytes. This hypothesis is now indicated in discussion section page 20 lines 5 to 11 as follows:

“Given that β-arrestin-mediated endocytosis contributes to cAMP-PKA-MAPK transduction pathway activation by many GPCRs (36,37), the inhibitory effect of Barbadine on β-arrestin receptor/complex clustering most likely contributes to the blockade of ODN metabotropic receptor-mediated PKA/MAPK-ERK1/2 activation in astrocytes, which could lead to the abrogation of the protective effect of ODN following Barbadin treatment”

12) The ability of ODN to prevent damage and signaling when given 48 hours after 6-OHDA is remarkable.  Can the authors mention the normal timeline of the 6-OHDA-induced damage in this system and when different signals are getting turned on/off after 6-OHDA addition?

Author Response

Previous in vitro experiments have demonstrated that ODN exerts a protective action against oxidative stress on glial cells (astrocytes) (Hamdi et al., 2012 PLoS One 7:e42498; Kaddour et al., 2019 Journal of Neurochemistry 125:620‒633). The protective effects of ODN is mediated through its metabotropic receptor, which rapidly activates (within the first 10 min after ODN administration) a transduction cascade of second messengers to stimulate protein kinase A (PKA), protein kinase C (PKC) and mitogen-activated protein kinase (MAPK)-extracellular signal-regulated kinase (ERK) signaling pathways, which in turn inhibits the expression of proapoptotic factor Bax and the mitochondrial apoptotic pathway. We have also revealed that ODN provokes a rapid and transient stimulation of SOD and catalase activities with a maximal effect occurring respectively 10 and 20 min after the onset of treatment.

In the present experiment, we cannot determine the exact onset of action of ODN, but in the light of our previous results, we hypothesize that exposure of ODN to 6-OHDA-treated cells induces rapid activation of the metabotropic ODN receptor and activation of the intracellular mechanisms responsible for inhibition of oxidative stress, neuroinflammation and toxin-induced apoptosis.

Round 2

Reviewer 1 Report

Comments and Suggestions for Authors

The authors must provide the original images of the western blots. The image in the revised manuscript is different from the image in the attached file named Original images for blots/gels, which present all the figures but not the images of the whole blot. 

Author Response

(The authors gave the same response as above.)

Round 3

Reviewer 1 Report

Comments and Suggestions for Authors

-